# High PD-L1/CD274 Expression of Monocytes and Blood Dendritic Cells Is a Risk Factor in Lung Cancer Patients Undergoing Treatment with PD1 Inhibitor Therapy

**DOI:** 10.3390/cancers12102966

**Published:** 2020-10-13

**Authors:** Dagmar Riemann, Wolfgang Schütte, Steffi Turzer, Barbara Seliger, Miriam Möller

**Affiliations:** 1Institute of Medical Immunology, Martin Luther University Halle-Wittenberg, 06112 Halle, Germany; steffi.turzer@uk-halle.de (S.T.); barbara.seliger@uk-halle.de (B.S.); 2Clinic of Internal Medicine, Hospital Martha-Maria Halle-Dölau, 06120 Halle, Germany; Wolfgang.Schuette@Martha-Maria.de (W.S.); Miriam.Moeller@Martha-Maria.de (M.M.)

**Keywords:** PD-L1/CD274, PD1 inhibitor therapy, lung cancer, flow cytometry, immune monitoring, dendritic cells, blood monocytes, CD16^+^ monocytes, survival analysis

## Abstract

**Simple Summary:**

Tumor cells can evade destruction via immune cells by expressing coinhibitory membrane molecules, which suppress antitumoral immune responses. Immune checkpoint inhibitor therapy acts by blocking these inhibitory pathways. Although this type of immunotherapy has shown promising results for selected cancer patients during recent years, an important challenge remains to identify baseline characteristics of patients who will mostly benefit from such therapy. The aim of our study was to assess the expression of the coinhibitory molecule PD-L1/CD274 on different antigen-presenting cells (monocytes and dendritic cell subsets) in the peripheral blood of 35 patients with nonsmall cell lung cancer undergoing checkpoint inhibitor therapy. CD274 expression correlated with therapy response and the survival of patients. Tumor patients with high CD274 expression levels of antigen-presenting cells in blood rarely responded to checkpoint inhibitor therapy. Our results implicate that a high CD274 expression in monocytes and dendritic cell subsets is a risk factor for therapy response.

**Abstract:**

The aim of this study was to investigate the expression of the coinhibitory molecule PD-L1/CD274 in monocytes and dendritic cells (DC) in the blood of lung cancer patients undergoing PD1 inhibitor therapy and to correlate data with patient’s outcome. PD-L1/CD274 expression of monocytes, CD1c^+^ myeloid DC (mDC) and CD303^+^ plasmacytoid DC (pDC) was determined by flow cytometry in peripheral blood at immunotherapy onset. The predictive value of the PD-L1/CD274-expression data was determined by patients’ survival analysis. Patients with a high PD-L1/CD274 expression of monocytes and blood DC subpopulations rarely responded to PD1 inhibitor therapy. Low PD-L1/CD274 expression of monocytes and DC correlated with prolonged progression-free survival (PFS) as well as overall survival (OS). The highest PD-L1/CD274 expression was found in CD14^+^HLA-DR^++^CD16^+^ intermediate monocytes. Whereas the PD-L1/CD274 expression of monocytes and DC showed a strong positive correlation, only the PD-L1/CD274 expression of DC inversely correlated with DC amounts and lymphocyte counts in peripheral blood. Our results implicate that a high PD-L1/CD274 expression of blood monocytes and DC subtypes is a risk factor for therapy response and for the survival of lung cancer patients undergoing PD1 inhibitor therapy.

## 1. Introduction

Promising benefits of immunotherapy, in particular those targeting the immune checkpoint proteins PD1 and PD-L1, have been shown in lung cancer patients in recent studies. Immune checkpoints are proteins that restrict physiologic immune cell responses in order to maintain immune homeostasis and protect host tissues from unnecessary damage due to excessive inflammation. Programmed cell death 1 ligand 1 (PD-L1), also known as B7-H1 and CD274, is a transmembrane protein expressed on the surface of antigen-presenting cells [1]. After binding to its cognate receptor PD1/CD279 on T cells, PD-L1/CD274 exerts regulatory actions via a negative costimulatory effect on T cell functions to inhibit cytokine secretion, facilitate apoptosis of activated T cells and induce T-cell anergy [1]. Since many tumors can express PD-L1/CD274, the rationale of the PD-L1 pathway blockade is to inhibit the immunosuppressive PD-L1/PD1 interaction between tumor cells and T cells that hampers the activity of CD4^+^ and CD8^+^ T cells thereby enhancing T cell-mediated antitumor activities [2,3]. Selected patients with advanced non-small cell lung cancer (NSCLC) profit from the treatment with the PD1 inhibitors pembrolizumab or nivolumab in first-or second-line settings. However, treatment with immune checkpoint inhibitors is associated with a unique pattern of immune-related adverse effects [4]. Furthermore, durable responses are only observed in a minority of patients and primary, adaptive and acquired therapy resistances are common [4,5,6]. 

An important challenge remains to identify the baseline characteristics of patients who will mostly benefit from immunotherapy treatment. Multicolor flow cytometry represents a powerful tool to characterize individual cells within heterogeneous cell populations. Our recent results of the characterization of blood immune cells in lung cancer patients undergoing checkpoint blockade therapy showed a poor survival for patients with a high neutrophil-to-lymphocyte-ratio (NLR), a high amount of HLA-DR^low^ monocytes and a low frequency of dendritic cells (DC) [7]. Since the PD-L1/CD274 expression of antigen-presenting cells might lead to an inhibition of antitumor responses following the presentation of tumor antigens to T cells, the aim of this study was to evaluate PD-L1/CD274 expression of blood monocytes and DC subpopulations in lung cancer patients undergoing PD1 inhibitor therapy with respect to their effect on therapy response and prognosis. 

## 2. Results

Table 1 shows the detailed characteristics of the 35 NSCLC patients of this study who received at least two cycles of PD1 inhibitor therapy. Pembrolizumab was offered to 18 of the 35 patients (51%), in seven cases (39%) as first-line- and in 11 (61%) as a second-line treatment. The remaining 17/35 (49%) patients received nivolumab. The mean follow-up time was 9.7 months (1–26 months) at the time of the data cut-off. The nine patients who stopped immunotherapy before the third cycle experienced a clinical worsening in most cases. Seven patients continued immune checkpoint inhibitor therapy. The tumor objective response rate was 40% for all patients with a median overall survival (OS) of 6.0 months and a 95% confidence interval (CI) of 3.2–8.8 months.

Table 2 summarizes the initial counts of monocytes, lymphocytes and blood DC subtypes as well as the CD274 expression of monocytes and DC subtypes in patients experiencing a clinical response (stable disease or remission) or a tumor progression. Between therapy responders and nonresponders, we did not observe differences between the pretherapeutic counts of monocytes and lymphocytes, respectively. Additionally, the amount of CD14^+^CD16^+^ monocytes and CD14^+^HLA-DR^++^CD16^+^ intermediate monocytes did not reveal significant differences, though with a high standard deviation. However, the higher the number of blood DC, the better the patient responded to therapy. 

Human blood DC are a rare heterogeneous cell population that comprise approximately 1% of peripheral blood mononuclear cells. DC are broadly defined as antigen-presenting cells with a high expression of MHC class II molecules that lack other leukocyte lineage markers (CD3, CD14, CD19 and CD56) [8]. With respect to their lineage origin, they can be classified into two major subsets: plasmacytoid DC (pDC) as the major producers of type-I interferon (IFN), and myeloid lineage DC (mDC). Based on their expression of CD1c and CD141, two further mDC subsets have been described [9] and were investigated in this study. Most patients with advanced lung cancer had very low amounts of blood DC, with the lowest values observed for CD141^+^ mDC. Despite measuring >10 E6 leukocytes by flow cytometry, often only <100 events of CD141^+^ mDC could be detected. Due to the poor statistics, we focused on pDC and CD1c^+^ mDC in our further investigations. pDC counts were especially low in nonresponders, with 2.6 ± 1.2 cells/μL in progressors with progression-free survival (PFS) ≤1 month, 6.9 ± 3.8 cells/μL in progressors with PFS >1 month, 9.0 ± 5.3 cells/μL in stable disease and 12.1 ± 7.0 cells/μL in remission. In addition, the percentage of CD1c^+^ mDC showed significantly higher values in therapy responders (Table 2). 

With respect to PD-L1/CD274 expression, monocytes had slightly higher intensities than DC subtypes (Table 2). Within the monocytic population, the proportion of CD14^+^HLA-DR^++^CD16^+^ intermediate monocytes (8.0 ± 4.1% of monocytes) had the highest PD-L1/CD274 expression (mean fluorescence intensity (MFI) of 1179 ± 660). Otherwise, no difference could be observed in the PD-L1/CD274 expression between CD1c^+^ mDC and pDC (Table 2, Figure 1). 

A high PD-L1/CD274 expression of monocytes and of DC subtypes was associated with a poor response to therapy. In patients responding to therapy compared to patients with progression, all the monocytic subgroups had a significantly lower PD-L1/CD274 expression (Figure 1). With respect to pDC, the percentage of PD-L1/CD274^+^ pDC was 31.9 ± 20.4 in patients with PFS ≤1 month, 21.1 ± 12.2 in progressors with a PFS >1 month and 12.5 ± 11.0 for patients with a clinical response. ROC analysis resulted in AUC values >0.700 (Table 2). With the cut-off points estimated by the Youden index method, univariate Kaplan–Meier and Cox regression analyses were performed for both PFS and OS, as given in Table 3. 

Patients with a higher PD-L1/CD274 expression of monocytes and DC subtypes, respectively, showed a significantly poorer survival. Figure 2 illustrates that patients with an initial value of >7.01 pDC/μL blood, ≤16% CD274^+^ pDC, a monocytic CD274 intensity of <480 and a CD274 intensity of CD1c^+^ mDC ≤450 had a significantly longer PFS.

A positive correlation between the PD-L1/CD274 intensities of monocytes and DC subtypes was observed (Table 4). In contrast, no correlation of the PD-L1/CD274 intensities of monocytes and DC, respectively, with the PD-L1/CD274 tumor status (provided by the Department of Pathology) was observed, though this value was already evaluated at the time point of histotype assignment. Furthermore, a high PD-L1/CD274 expression of pDC significantly correlated with low amounts of both pDC and CD1^+^ mDC (Table 4). Furthermore, the percentage of PD-L1/CD274^+^ pDC inversely correlated with the number of blood lymphocytes, with similar results for T cells, B cells and NK cells. In contrast, monocytic PD-L1/CD274 intensity did neither correlate with the amount of pDC or CD1c^+^ mDC, nor with lymphocyte counts. 

## 3. Discussion

Checkpoint inhibition has complemented the therapeutic approach for patients with advanced lung cancer, although not all the patients benefited from it [4]. Understanding the reasons for patients’ variability in response to therapy and developing reliable biomarkers to predict patients, who are likely to respond to therapy, remains a challenge. PD-L1/CD274 expression in tumor tissues has emerged as one such candidate biomarker of therapy response, since patients with PD-L1/CD274-expressing advanced tumors have a higher objective response rate and improved PFS and OS as compared to the negative subgroups [10]. In NSCLC, the positive prognostic value of PD-L1/CD274 expression was independent of age, stage and histotype [11]. 

The primary rationale of checkpoint blockade therapy was to inhibit the immunosuppressive PD-L1/PD1 interaction between tumor cells and T cells that hampers the activity of CD4^+^ and CD8^+^ T cells [1]. However, in recent studies PD-L1/CD274 expression by tumor tissues was associated with the presence of tumor-infiltrating lymphocytes, which could be involved in better immunotherapy-triggered prognosis [12,13]. PD-L1/CD274 is expressed at low levels on a wide range of cells and its expression can be upregulated in response to various stimuli (review in [14]). In the context of tumor microenvironments, cells including macrophages, DC, myeloid-derived suppressor cells, regulatory T cells and endothelial cells can upregulate PD-L1/CD274 due to inflammation responses. Since the primary function of coinhibitory receptor/ligand pairs is to attenuate the magnitude and duration of immune responses in order to minimize collateral tissue damage during a host immune response, PD-L1/CD274 expression of antigen-presenting cells might contribute to tumor escape. 

In this study, PD-L1/CD274 expression of monocytes and blood DC subtypes in NSCLC patients undergoing PD1 inhibitor therapy was investigated. A high expression of this molecule was found to be a poor prognostic factor. Our results are in contrast to data from murine tumor models, where PD-L1/CD274-expressing antigen-presenting cells, rather than tumor cells, demonstrated essential antitumor effects of anti-PD-L1 monotherapy. A positive response to checkpoint inhibitor therapy has been associated with a high expression of PD-L1/CD274 on tumor-infiltrating immune cells indicating a role for PD-L1/CD274-expressing immune cells in suppressing antitumor responses, which are reinvigorated on checkpoint blockade therapy [15]. However, PD-L1/CD274-expressing monocytes and blood DC kept a significant negative impact on prognosis in this study. One could postulate that the onset of checkpoint inhibitor therapy was too late to reverse the pronounced immune suppression demonstrable in some of the NSCLC patients with advanced tumor stages. Adenocarcinoma was the most common tumor type in this study and an aggressive and early progressing nonsquamous NSCLC has been described, which might even represent a distinct disease entity [16]. Otherwise, the functional consequences of a PD-L1/CD274 expression could be affected by binding partners or molecules coexpressed with this molecule on antigen-presenting cells. Furthermore, besides PD1, PD-L1/CD274 can also bind to CD80 on activated T cells, thereby delivering another inhibitory signal [17,18], which is not inhibited by anti-PD1 antibody therapy. CD80 has been shown to interact with PD-L1/CD274 in cis on antigen-presenting cells to disrupt PD-L1/PD1 binding [19], and CD80 expression might differ between antigen-presenting cells in blood and tumor tissue. Furthermore, factors mediating PD-L1/CD274 expression of blood immune cells might exert pleiotropic immunosuppressive functions. These include for example immunosuppressive cytokines, such as IL-10 and IL-27, as well as the activation of different (oncogenic) signal transduction pathways, such as myc and phosphatidylinositol 3-kinase/Akt [20,21].

We observed that the percentage of PD-L1/CD274^+^ pDC inversely correlated with lymphocyte counts and pDC numbers. Very low amounts of DC were found in some of the patients with advanced lung cancer, which might contribute to the disturbed immune functions and poor prognosis. Several tumor-derived factors could be responsible for the decline of DC, e.g., increased serum levels of IL-10 correlate with profound numerical deficiency and immature phenotype of circulating DC subsets in patients with hepatocellular carcinoma [22]. NSCLC patients with low pretherapeutic values of blood pDC had a poor therapy response. In settings of cancer, pDC-derived type-I IFNs can promote antitumoral immunity through their direct activity on both tumor and immune cells [23]. pDC secrete a multitude of other inflammatory cytokines and chemokines and can act as antigen-presenting cells, although with lower efficacy than conventional DC [24]. Our earlier results in lung cancer patients showed that blood DC numbers decrease with age and tumor stage [25]. In addition, an increase of blood DC levels could be found in such patients, which did respond to checkpoint inhibitor therapy [7]. 

Tumors develop multiple strategies that lead to immune suppression thereby preventing effective antitumor immunity, such as the increased secretion of immunosuppressive metabolites and cytokines, e.g., IL-10 and TGF-β, enhanced differentiation of immune effector cells to a regulatory phenotype, as well as an accumulation of immunosuppressive cells, such as myeloid-derived suppressor cells [3]. Depending on the signals received from the microenvironment, DC can either activate adaptive immune responses or mediate immune tolerance. Immunogenic DC are characterized by a high expression of costimulatory molecules and the production of proinflammatory cytokines, whereas tolerogenic DC express low levels of costimulatory molecules and produce immunomodulatory cytokines. DC treated with lung cancer cell culture supernatants significantly downregulated the expression of MHC class II molecules and of the costimulatory molecules CD40 and CD80, but upregulated the inhibitory molecule PD-L1/CD274 [26]. Signals generated from inhibitory checkpoint molecules might contribute to the inhibitory properties of DC in cancer patients. Furthermore, PD-L1/CD274 silencing on DC could enhance T-cell responses leading to tumor clearance [2], which is in accordance with several studies demonstrating the advantages of knocking down PD-L1/CD274 regarding the efficacy of DC vaccine therapy [27,28]. Whereas a negligible PD-L1/CD274 expression of blood pDC and mDC of healthy donors has been described [29], blood DC of lung cancer patients show a clear PD-L1/CD274 expression in this study, thereby confirming the data of blood DC in patients with ovarian cancer [2] and melanoma [30]. Similarly, monocytes in healthy controls express only a small amount of PD-L1/CD274, whereas monocytes in cervical cancer patients show an increased expression [31]. Our results show that PD-L1/CD274 expression of monocytes and DC was positively correlated suggesting common ways of regulation in the different cell types. PD-L1/CD274 expression can be upregulated by a substantial number of mediators (for a review see [14,20,32]). As an example, PD-L1/CD274 expression of monocytes and DC has been found upregulated in response to the presence of T cells producing immune-stimulating cytokines, such as IFNs [33,34]. Other known inductors on monocytes and/or DC are IL-17 [35], TNF-α [36], IL-10 [20] and TGF-β [37]. Human blood contains several forms of soluble or extracellular PD-L1, included, e.g., in exosomes and microvesicles [38], which could be involved in the induction of PD-L1/CD274 expression on antigen-presenting cells. In advanced NSCLC, high levels of soluble PD-L1/CD274 correlated with nivolumab treatment failure [39], and serum with a high proportion of PD-L1/CD274^+^ exosomes, have been shown to inhibit in vitro IL-2 and IFN-γ production by CD8^+^ T cells [40].

Monocytic PD-L1/CD274 expression, which was also a poor risk factor in our study, did neither correlate with lymphocytic nor with blood DC counts. Our results show that CD14^+^CD16^+^ intermediate monocytes with a high HLA-DR intensity expressed the highest PD-L1/CD274 levels. Monocytes egress from the bone marrow as a uniform population of CD14^+^CD16-negative cells, a proportion of which subsequently differentiates to become intermediate (CD14^+^CD16^+^ and high amounts of HLA-DR) and “non-classical monocytes” monocytes (CD14dimCD16^+^) [41]. Intermediate monocytes show a high phagocytic activity and produce IL-10 [42]. Since IL-10 is known to inhibit HLA-DR expression [43] and the intermediate monocytes express a rather high HLA-DR intensity, IL-10 might not be the responsible inductor of monocytic PD-L1/CD274 expression. Future investigation will show whether IL-10 is involved in the high PD-L1/CD274 expression of monocytes and blood DC subtypes observed in lung cancer patients with a poor therapy response to anti-PD1 therapies. 

Although the efficacy of immune checkpoint inhibitors is well-established in oncology, there is increasing evidence that their use may also be effective in several noncancer acute and chronic inflammatory conditions, including sepsis, burns and chronic infections [44]. An increased frequency of PD-L1/CD274-expressing monocytes is an independent risk factor for infectious complications in acute pancreatitis [45]. In sepsis, high monocytic PD-L1/CD274 expression has been correlated with increased T-cell apoptosis, lymphopenia, and T-cell dysfunction [46]. Whereas PD1 expression on T cells was not a reliable “danger signal” for immune suppression in septic patients, monocytic PD-L1/CD274 intensity was an independent predictor of 28-day mortality in septic shock patients [47]. PD-L1/CD274 expression of CD14^+^CD16^+^ intermediate monocytes has been described upon hepatitis C virus (HCV) infection. The upregulation of monocytic PD-L1/CD274 intensity was associated with defective HCV-specific T-cell responses, while the inhibition of monocyte-associated PD-L1/CD274 expression enhanced the frequency of IFN-γ-producing HCV-specific T cells and the production of Th1 cytokines [48]. PD-L1/CD274 expression of DC was also increased in HCV-infected patients and this increase was associated with an impaired allostimulatory capacity of DC [49].

Despite notable and durable clinical responses, basic and clinical studies are still required to determine the exact mechanism of checkpoint inhibitor immunotherapy, and the appropriate selection of patients. Currently, major efforts are being made to elucidate the mechanisms involved in the development of primary and acquired resistance to checkpoint inhibitor therapy [50]. By understanding the resistance mechanisms involved, strategies can be designed to overcome resistance and treatment failure. PD-L1/CD274 expression of monocytes and blood DC could be involved in cancer-induced immune suppression and can be used as a blood biomarker for poor response to PD1 inhibitor therapy. The factors responsible for PD-L1/CD274 expression of blood immune cells, as well as the role of PD-L1/CD274-expressing CD14^+^HLA-DR^++^CD16^+^ intermediate monocytes, needs to be clarified in future investigations. Considering that cancer immunotherapy is the most actively evolving therapy for lung cancer, we believe that this study has some important findings, which should be further pursued by confirmatory and extended studies. 

## 4. Materials and Methods

### 4.1. Patient Cohort

The institutional review board of the Ärztekammer Sachsen-Anhalt approved this study (No. 96018). A 2.7 mL volume peripheral blood was collected from the 35 prospectively enrolled patients with the criteria: histologically confirmed diagnosis of metastatic NSCLC, age >18 years, adequate organ functions, medical decision-making capacity, available PD-L1/CD274 status determined by immunohistochemical analysis, epidermal growth factor receptor (EGFR) wild-type, negative for anaplastic lymphoma kinase (ALK) translocation, no previous history of systemic immunosuppressive therapy and no active autoimmune disease. Patients enrolled received either pembrolizumab as monotherapy (200 mg in chemotherapy-naïve patients, 2 mg/kg for patients previously treated with chemotherapy, every 3 weeks) or nivolumab intravenously administered (3 mg/kg every 2 weeks). PD-L1 tumor status and patients’ treatment history determined the choice of agent (first-line or second-line setting). Every 9 weeks or with clinical worsening of the patient’s condition, scheduled computed tomography (CT) or magnetic resonance imaging was performed. A treatment benefit was defined as complete/partial remission, and stable disease on CT scan according to RECIST 1.1. Patients with progressive disease at the first CT scan were categorized as no disease control. Treatment continued until confirmed disease progression, unacceptable toxicity, or withdrawal of consent.

### 4.2. Antibody Staining and Flow Cytometry

Peripheral blood from NSCLC patients was taken on the day of immunotherapy start. At first, the complete leukocyte blood count was monitored, then antibody staining of whole blood was performed. The “Blood DC Enumeration Kit” of company Miltenyi Biotec (Bergisch Gladbach, Germany) was supplemented with the monoclonal antibody (mAb) CD16 for the detection of CD16^+^ monocytes, and with an HLA-DR mAb for better gating possibilities. In brief, whole blood samples were labeled with the mAbs CD303 FITC as a pDC marker [51], CD1c phycoerythrin (PE) for mDC (conventional DC2), CD14/CD19 PE-Cy5, CD141 allophycocyanin (APC) for mDC (cDC1), CD16 PE-Cy-7 (BioLegend, Koblenz, Germany), HLA-DR V500 and CD274 BV421 (BD Biosciences, Heidelberg, Germany). According to the manufacturer’s instruction, mAb incubation, red cell lysis, two washing steps and cell fixation were performed. Samples were measured with a FACS CANTO II flow cytometer (BD Biosciences) with FACS DIVA^TM^ software. To set standardized geometric MFI ranges in the fluorescence channels used, Cytometer Setup and Tracking Beads (BD Biosciences) were used daily. At least 1 × 10^6^ blood leukocytes were analyzed. The gating strategy for DC subpopulations was described earlier [7]. Monocytes were gated in a CD14/SSC plot, and a CD16/HLA-DR plot was used to identify CD14^+^CD16^+^ and CD14^+^HLA-DR^++^CD16^+^ monocytes (Figure 1). BV421 histograms were used to estimate the CD274 MFI and the percentage of CD274-positive DC, with CD274 staining of B cells serving as a control

### 4.3. Statistical Analysis

The commercial software SPSS 25.0 (SPSS Inc., Munich, Germany) was used for all statistical analyses. ANOVA analysis and a Student’s *t*-test were used to investigate the differences in immune cell numbers between responders and nonresponders to therapy. All *p*-values are exploratory. Spearman correlation coefficients (CC) were calculated to investigate correlations between PD-L1/CD274 expression of monocytes and DC, respectively, with immune cell parameters. For survival analysis, PFS was defined as the time from the first PD1 inhibitor treatment to tumor progression or death, OS was the duration of survival after starting immunotherapy. Survival analysis firstly comprised a descriptive representation of the cumulative survival functions according to Kaplan-Meier. The log-rank test was used to identify differences among the survival curves. To examine the correlation of immune cell values with PFS and OS, Cox regression analysis was performed. *p* < 0.05 was considered statistically significant. Predictor variables with a significant difference between patients’ groups with and without a therapy response were analyzed with receiver operating characteristics (ROC) curves to determine the overall strength of association (area under the ROC curve (AUC)), as well as the optimal cut-point for the prediction of therapy response. Youden’s Index was used to calculate which cut-off point gives the best sensitivity and specificity (with J = sensitivity + specificity-1).

## 5. Conclusions

The observed heterogeneity in clinical responses to checkpoint blockade therapy in cancer patients has led to major efforts to define biomarkers predicting therapy responses. Using multicolor flow cytometry, we prospectively monitored blood immune cells from patients with advanced NSCLC undergoing therapy with PD1 inhibitors to investigate the consequences of a PD-L1/CD274 expression of monocytes and DC in peripheral blood. A high pretherapeutic PD-L1/CD274 expression has been detected as an adverse factor for PD1 inhibitor therapy in this study, with the highest PD-L1/CD274 expression found in CD14^+^HLA-DR^++^CD16^+^ intermediate monocytes. Since PD-L1/CD274-expressing monocytes and blood DC may be of pathophysiological relevance, a better understanding of the underlying mechanisms of their regulation and the functional consequences of a PD-L1/CD274 expression on blood immune cells might help to generate novel hypotheses for immune evasion and might offer novel opportunities for the design and optimization of immunotherapies. Rather than assessing only the PD-L1/CD274 expression of tumor cells, the additional monitoring of PD-L1/CD274 expression of immune cells in the blood appears to be mandatory for predicting therapy responses in patients undergoing checkpoint blockade therapy. 

## Figures and Tables

**Figure 1 cancers-12-02966-f001:**
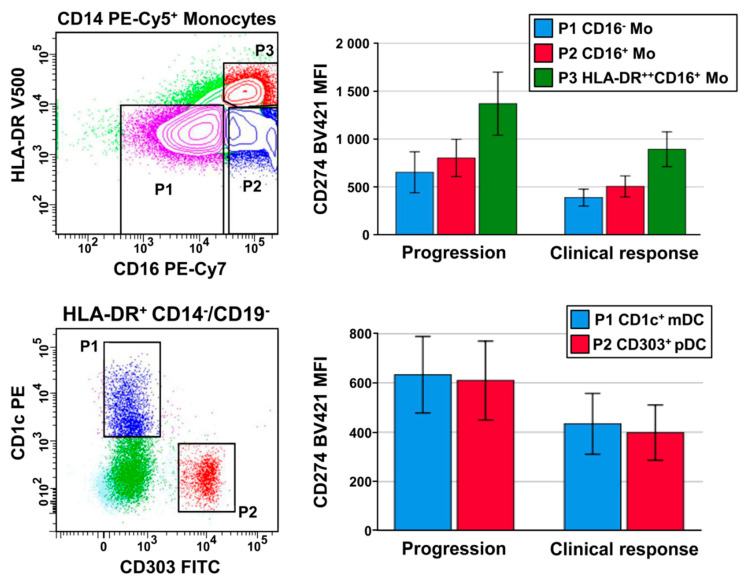
Gating strategy and PD-L1/CD274 mean fluorescence intensity (MFI) for CD14^+^CD16-negative classical monocytes, CD14^+^CD16^+^ monocytes and CD14^+^HLA-DR^++^CD16^+^ intermediate monocytes (upper part), and for CD1c^+^ myeloid DC (mDC) and CD303^+^ plasmacytoid DC (pDC) (lower part of the picture). Bars illustrate mean value and standard error with a significant difference between the outcome “progression” and “clinical response” with respect to the PD-L1/CD274 intensity of CD14^+^CD16-negative monocytes (*p* = 0.029), CD14^+^CD16^+^ monocytes (*p* = 0.034), CD14^+^HLA-DR^++^CD16^+^ monocytes (*p* = 0.027), CD1c^+^ mDC (*p* = 0.041) and pDC (*p* = 0.042).

**Figure 2 cancers-12-02966-f002:**
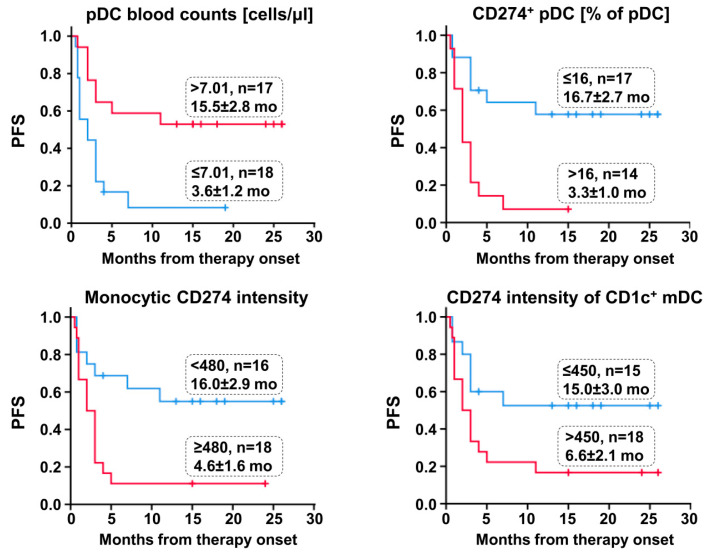
Kaplan–Meier curves showing progression-free survival (PFS) for patients undergoing PD1 inhibitor therapy and categorized with pDC blood counts, the percentage of CD274^+^ pDC, monocytic CD274 expression (mean fluorescence intensity (MFI)) and CD274 expression of CD1c^+^ mDC (MFI). Tick marks indicate censored observations. Cut-off point, patient number (*n*) and the mean ± standard error of the estimated PFS are given for each group. Survival statistics are shown in Table 3.

**Table 1 cancers-12-02966-t001:** Patient characteristics.

Parameters	Characteristics	*N* (%)
Age at start of immunotherapy, years *n* (%)	Median	65
	Range	24–85
	>75 years	6 (17)
Sex, *n* (%)	Male Female	19 (54) 16 (46)
Histology, *n* (%)	Adenocarcinoma Squamous cell carcinoma mixed	23 (66) 7 (20) 5 (14)
Smoking status	Current or former smokers Never smokers	30 (86) 5 (14)
PD-L1/CD274 tumor expression, *n* (%)	<1% ≥1–49% >49 Missing	11 (31) 9 (26) 14 (40) 1
Response, *n* (%)	Stop after 2 treatment cycles Progressive disease after ≥3 cycles disease stabilization Partial remission	9 (25.7) 12 (34.3) 7 (20.0) 7 (20.0)

**Table 2 cancers-12-02966-t002:** Pretherapeutic counts of monocytes, lymphocytes and blood dendritic cells (DC) subtypes as well as programmed cell death 1 ligand 1 (PD-L1)/CD274 expression of monocytes and DC subtypes in the patients’ groups “clinical response” (*n* = 14) and “progression” (*n* = 20). The *p*-value of the Student’s *t*-test, the area under the ROC curve (AUC) showing the discrimination capability of the marker with respect to progression-free survival (PFS), as well as the cut-point value (Youden index method), are shown (MFI, mean fluorescence intensity).

Immune Cell Subtypes	Clinical Response	Progression	*p*-Value	AUC	Cut-off Value
Leukocytes (cells/μL)	8597 ± 2262	9600 ± 3175			
Neutrophils (cells/μL)	6214 ± 1948	7326 ± 3140			
Monocyte counts (cells/μL)	626 ± 160	672 ± 261			
CD14^+^CD16^+^ monocytes (% of monocytes)	23.6 ± 19.3	16.4 ± 11.6			
CD14^+^HLA-DR^++^CD16^+^ monocytes (% of monocytes)	8.3 ± 3.8	7.7 ± 4.3			
Lymphocytes (cells/μL)	1459 ± 520	1413 ± 628			
CD303^+^ pDC counts (cells/μL)	10.6 ± 6.2	5.9 ± 3.9	0.009	0.745	7.01
CD303^+^ pDC (% of leukocytes)	0.119 ± 0.054	0.070 ± 0.050	0.011	0.769	0.061
CD1c^+^ mDC (cells/μL)	13.2 ± 8.5	9.3 ± 8.6			
CD1c^+^ mDC (% of leukocytes)	0.146 ± 0.068	0.089 ± 0.064	0.018	0.755	0.104
CD141^+^ mDC (% of leukocytes)	0.0122 ± 0.009	0.006 ± 0.001	0.019		
Monocytic CD274 intensity (MFI)	450 ± 180	757 ± 468	0.027	0.750	480
CD274 intensity of pDC (MFI)	398 ± 194	609 ± 331	0.042	0.722	440
CD274^+^ pDC (% of pDC)	12.5 ± 11.0	24.5 ± 15.4	0.022	0.730	16.0
CD274 intensity of CD1c^+^ mDC (MFI)	433 ± 214	633 ± 322	0.041	0.705	450
CD274^+^ mDC (% of CD1c^+^ mDC)	15.09 ± 13.77	26.0 ± 18.11	0.062		

**Table 3 cancers-12-02966-t003:** Relationship between initial pDC counts and PD-L1/CD274 expression of monocytes and DC subtypes, respectively, with patient’s progression-free survival (PFS) (**A**) and overall survival (OS) (**B**). Data of univariate prognostic factor analysis (Kaplan–Meier and Cox regression) are shown (HR, hazard ratio; CI, confidence interval; MFI, mean fluorescence intensity).

A	Cut-Off Value	*n*	Kaplan–Meier			Cox Regression		
			% Censored	PFS Time (Months)	Log-Rank Test	HR	95% CI	*p*-Value
Blood pDC counts (cells/μL)	≤7.0	18	11.1	3.65 ± 1.236	0.002	3.455	1.427–8.365	0.006
	>7.0	17	52.9	15.46 ± 2.76				
Monocytic CD274 expression (MFI)	<480	16	56.3	16.00 ± 2.87	0.007	3.116	1.242–7.814	0.015
	≥480	18	11.1	4.62 ± 1.64				
CD274 MFI of pDC	≤440	19	47.4	13.95 ± 2.65	0.026	2.414	1.029–5.660	0.043
	>440	14	14.3	5.23 ± 2.08				
CD274^+^ pDC (% of pDC)	≤16	17	58.8	16.66 ± 2.73	0.001	4.14	1589–10,784	0.004
	>16	14	7.1	3.32 ± 0.96				
CD274 MFI of CD1c^+^ mDC	<450	15	53.3	15.01 ± 3.04	0.031	2.464	0.997–6.086	0.051
	≥450	18	16.7	6.57 ± 2.12				
**B**	**Cut-Point**	***n***	**Kaplan–Meier**			**Cox Regression**		
			**% Censored**	**OS Time (Months)**	**Log-Rank** **Test**	**HR**	**95% CI**	***p-*** **Value**
Blood pDC counts (cells/μL)	≤7.0	18	11.1	5.94 ± 1.27	0.002	3.548	1.477–8.523	0.005
	>7.0	17	52.9	16.8 ± 2.44				
Monocytic CD274 expression (MFI)	<480	16	56.3	17.06 ± 2.61	0.004	3.343	1.334–8.376	0.010
	≥480	18	11.1	6.83 ± 1.63				
CD274 MFI of pDC	≤440	19	47.4	15.21 ± 2.41	0.028	2.397	1.024–5.607	0.044
	>440	14	14.3	7.57 ± 2.04				
CD274^+^ pDC (% of pDC)	≤16	17	58.8	17.78 ± 2.44	0.001	4.011	1.532–10.501	0.005
	>16	14	7.1	6.5 ± 1.54				
CD274 MFI of CD1c^+^ mDC	<450	15	53.3	16.33 ± 2.72	0.035	2.441	0.989–6.023	0.053
	≥448	18	16.7	8.61 ± 1.99				

**Table 4 cancers-12-02966-t004:** Association of PD-L1/CD274 expression of monocytes and DC subtypes with other immune cell markers, analyzed by Spearman’s rank correlation. Correlation coefficient (CC) and *p*-values are shown.

Correlation of	CC	*p*-Value
**Monocytic CD274 Expression with**		
CD274 expression of pDC	0.954	<0.001
CD274 expression of CD1c^+^ mDC	0.861	<0.001
**CD274^+^ pDC (% of pDC) with**		
Proportion of pDC (% of leukocytes)	–0.523	0.003
Proportion of CD1c^+^ mDC (% of leukocytes)	–0.416	0.022
Lymphocytes (cells/μL)	–0.632	<0.001

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
