# Peer review of "High PD-L1/CD274 Expression of Monocytes and Blood Dendritic Cells Is a Risk Factor in Lung Cancer Patients Undergoing Treatment with PD1 Inhibitor Therapy"

_cancers, 2020, doi:10.3390/cancers12102966_

Round 1

Reviewer 1 Report

  1. interesting and well conducted work, albeit in a limited series
  2. In the discussion: line 179-181 "our results...monotherapy": unnecessary reference to the descriptive context
  3. in the discussion: the part on biohumoral factors (IL / VEGF etc) is not congruent with what was done in the study so it could be omitted making the discussion more streamlined
  4. in the conclusions I would specify how this work can help in generating research hypotheses but to date it is not sufficient to transpose the results into the clinical setting

Author Response

Answer to reviewer 1

  1. interesting and well conducted work, albeit in a limited series
  2. In the discussion: line 179-181 "our results...monotherapy": unnecessary reference to the descriptive context

Answer:

In the sentence “our results…monotherapy”, the references have been discarded now.

  1. in the discussion: the part on biohumoral factors (IL / VEGF etc) is not congruent with what was done in the study so it could be omitted making the discussion more streamlined

Answer: As suggested by the reviewer, the sentence “Other mediators known…” in the discussion part on biohumoral factors declining DC numbers was deleted in the new version of the manuscript.

  1. in the conclusions I would specify how this work can help in generating research hypotheses but to date it is not sufficient to transpose the results into the clinical setting

Answer: The conclusion was revised in the new version of the manuscript. The part on “transposing results into clinical settings” was deleted. Instead, we propose that the characterization of blood cells could supplement monitoring of patients undergoing immunotherapy.

Reviewer 2 Report

While there are patients who do not show benefit to immune check point inhibitors the results of the manuscript is quite interesting.

Minor comment: Youden index mothos which was mentioned in the 'RESULT' should be explained in the 'METHOD' section.

Author Response

Answer to reviewer 2

While there are patients who do not show benefit to immune check point inhibitors the results of the manuscript is quite interesting.
Minor comment: Youden index mothos which was mentioned in the 'RESULT' should be explained in the 'METHOD' section.

Answer: The description of the method for "Youden’s Index" mentioned in “Results” is now included in section 4.3 of “Materials and Methods”.

Reviewer 3 Report

Overall nice paper with interesting results, that may help refine PD1 inhibitor therapy. The double statistical analysis in table 3, both Kaplan Meier and Cox regression is not necessary, but can be accepted. Figure 2 most clearly shows the interest of measuring CD274 expression and intensity on monocytes and DC.

Author Response

The reviewer is right, to show both Kaplan Meier and Cox regression is not necessary.
However, this calculation path was advised us by the university statisticians.
At first we used Kaplan-Meier method with log-rank test for comparing the survival curves in the two groups.
To get also the hazard ratio (probability of the endpoint) we continued with Cox regression.
We think that both methods complement each other.
In future studies, with more patients, we will calculate multiparametric Cox regression method.

Reviewer 4 Report

quite interesting result, which is in contrast with many similar animal models.

1) The sample number is not very big, raising some concerns about the robustness of the final conclusions. Can authors provide more data to compare, like normal blood DC PD-1 distribution? or similar inhibited situations?

2) "Furthermore, factors mediating PD-L1/CD274 expression of blood
immune cells might exert pleiotropic immunosuppressive functions." Can authors give more explanations and references on this?

3) "Monocytic PD-L1/CD274 expression, which was also a poor risk factor in our study, did neither correlate with lymphocytic nor with blood DC counts", this finding is kind of interesting, hard to relate to any molecular or kinetic mechanisms. One may doubt the sensitiveness of the detection approach. The authors may want to provide more hypothesis.

4) Would the conclusions stand for other inhibitors to expand their generality or we should execute with much caution in the immuno therapy in the future for such cases.

Author Response

Answer to reviewer 4

1) The sample number is not very big, raising some concerns about the robustness of the final conclusions. Can authors provide more data to compare, like normal blood DC PD-1 distribution? or similar inhibited situations?

Answer: The reviewer is right. As far as we know, this is the first patient cohort measuring CD274 expression of monocytes and blood DC during checkpoint blockade, and data have to be confirmed with more study patients in the future. We did not include an age-matched control population during our measurements. However, in the manuscript is cited a paper dealing with the expression of several co-inhibitory molecules on DC [Carenza C et al., Front Immunol 10 (2019)1325]. The authors investigated the expression of three immune checkpoints on peripheral blood DC subsets and found a low PD-L1 expression level on slan+DC and negligible expression in all other subsets. 
In the first version of the manuscript, we already cite data on a CD274 expression of blood DC in ovarian cancer and melanoma (page 9). With respect to monocytes, we now include a paper on CD274 expression of monocytes in cervical cancer patients, where a low CD274 expression of monocytes of healthy controls (20%) has been mentioned [Zhang Y et al., Oncol. Lett 14 (2017)].

In the first manuscript version we discuss monocytic CD274 expression as a risk factor in sepsis and chronic infections.

2) Furthermore, factors mediating PD-L1/CD274 expression of blood immune cells might exert pleiotropic immunosuppressive functions." Can authors give more explanations and references on this?

This includes for example immune suppressive cytokines, such as IL-10 and IL-27 as well as activation of different (oncogenic) signal transduction pathways, such as myc and phosphatidylinositol 3-kinase/Akt [20,21]. 
We supplemented this sentence in the manuscript, as suggested by the author. Furthermore, we cite reviews dealing with CD274 regulation (e.g., Sun C et al., Immunity 2018, 48, 434-452).

3) "Monocytic PD-L1/CD274 expression, which was also a poor risk factor in our study, did neither correlate with lymphocytic nor with blood DC counts", this finding is kind of interesting, hard to relate to any molecular or kinetic mechanisms. One may doubt the sensitiveness of the detection approach. The authors may want to provide more hypothesis.

We can exclude sensitivity problems, since the CD274 staining of monocytes and DC was performed in one tube, with one kind of CD274 antibody. 
To date, we do not have an explanation for the difference between CD274 expressing DC (high expression does correlate with low DC numbers and low lymphocyte counts) and CD274 expressing monocytes (high expression does not correlate with DC numbers, monocytes and lymphocyte counts). However, data with respect to correlating DC frequencies and lymphocyte counts are in line with our recent studies in lung cancer patients (Moller et al., J Immunother 2020, 43, 57-66). 
As mentioned in the manuscript, the functional consequences of a CD274 expression could be affected by binding partners or molecules co-expressed with CD274 on antigen-presenting cells. We will follow up on this issue in future studies.

4) Would the conclusions stand for other inhibitors to expand their generality or we should execute with much caution in the immuno therapy in the future for such cases.

Our results are restricted to CD274 expression of blood antigen-presenting cells. However, we are convinced that data can be extended to other co-inhibitory receptor/ligand-pairs, such as OX40 and OX40 ligand. Much more knowledge is needed to understand the mechanisms of the synergy between different immune checkpoints to successfully use combination therapies in future trials.